# Role of Epigenetics in Type 2 Diabetes and Obesity

**DOI:** 10.3390/biomedicines9080977

**Published:** 2021-08-08

**Authors:** Rosanna Capparelli, Domenico Iannelli

**Affiliations:** Department of Agriculture Sciences, University of Naples “Federico II”, Via Università, 100-Portici, 80055 Naples, Italy

**Keywords:** type 2 diabetes, obesity, epigenetic, genetics

## Abstract

Epigenetic marks the genome by DNA methylation, histone modification or non-coding RNAs. Epigenetic marks instruct cells to respond reversibly to environmental cues and keep the specific gene expression stable throughout life. In this review, we concentrate on DNA methylation, the mechanism often associated with transgenerational persistence and for this reason frequently used in the clinic. A large study that included data from 10,000 blood samples detected 187 methylated sites associated with body mass index (BMI). The same study demonstrates that altered methylation results from obesity (OB). In another study the combined genetic and epigenetic analysis allowed us to understand the mechanism associating hepatic insulin resistance and non-alcoholic disease in Type 2 Diabetes (T2D) patients. The study underlines the therapeutic potential of epigenetic studies. We also account for seemingly contradictory results associated with epigenetics.

## 1. Introduction

Epigenetics is often broadly defined and vaguely used [1]. According to the definition more frequently cited, epigenetics is the study of hereditable changes in gene function that occurs without altering the DNA sequence [2]. The view that epigenetic markers are transmitted to the next generation via the germline is not generally accepted: first, because epigenetic changes have not yet been detected in germ cells [3]; second, because in mammalian germ cells and somatic cells are separated by the Weisman barrier [3]. At present, the hypothesis prevails that transmission of epigenetic changes occurs through long-lived RNA molecules that can pass through the Weisman barrier [4]. Epigenetics marks the genome by DNA methylation, histone modification, or non-coding RNAs. Epigenetic marks instruct cells to respond reversibly to environmental cues and—at the same time—keep the specific gene expression programs stable throughout life [5]. Estimates of hereditability of complex diseases (such as OB or T2D) generally account for a small fraction of the genetic variability [6]. We can explain this conclusion by pointing out that many of these studies have ignored the epigenome. To better understand the phenotypic differences characterizing single patients, mapping the epigenome may be as informative as mapping the genome [6].

Clinical epigenetics is a promising approach for the diagnosis of several complex diseases, including OB and T2D [7]. Specifically, the detection of epigenetic biomarkers represents an appealing practice, both in clinical and research. In this regard, sophisticated techniques are rapidly evolving. To study histone modifications or chromatin conformation, are commonly used chromatin immunoprecipitation (ChIP) or chromatin conformation capture (3C)-derived assay [8]. Instead, quantitative measurement of DNA methylation (requiring DNA conversion by bisulfite or DNA digestion by methylation-specific restriction endonucleases) [9] represents a valid method to analyze DNA methylation profiles. However, limitations in the use of epigenetic biomarkers remain, due to epigenetic plasticity [7].

The study of epigenetic traits requires the distinction between intergenerational and transgenerational inheritance [10]. Intergenerational epigenetic traits occur when the exposure to the maternal womb (F0) influences the developing fetus and its germline, leading to an altered phenotype of the child (F1) and possibly, of the grandchild (F2); exposure to the paternal environment can affect the germ cells that will generate the child (F1). Instead, a true transgenerational epigenetic trait is transmitted across generations: to the F2 generation in case of exposure to the paternal line, or to the maternal line if exposure occurs before conception; to the F3 and the following generations, if exposure occurs during pregnancy and in absence of a new environmental exposure or germline mutations (Figure 1).

For a long time, high levels of methylation were associated with gene silencing. It is now known that generally DNA methylation of promoters or enhancers is associated with gene silencing, while methylation in the rest of the gene is associated with active gene expression [11]. Finally, in the majority of the methylation studies, level and phenotype (such as T2D or OB) are measured at the same time. This procedure is a major obstacle to ascertain whether DNA methylation is the cause or effect of the disease [11].

Epigenetic changes occur early in life and pass on to the next generation prevalently via intergenerational inheritance [12]. Early expression of epigenetic marks contributes to keeping constant the gene expression patterns of distinct cell types. During the intrauterine life, an excess or scarcity of nutrients may induce epigenetic changes in the child and increase the risk of OB, T2D, or cardiovascular disease (CVD) in the adult [13,14,15,16,17]. How an excess or scarcity of nutrients may induce epigenetic changes, at present is still not clear. Nutritional reprogramming of neural, endocrine or metabolic cells can be potential causal mechanisms [18].

Animal studies have demonstrated that insufficient nutrition during intrauterine life induces epigenetic changes in the offspring [19]. These results agree with a study carried out on children whose mothers—during World War II—suffered prolonged food deprivation. The children—once adults—displayed reduced methylation of the gene *IGF2* [20], OB or glucose intolerance, depending upon the length of starvation [21].

This study is a review of epigenetics in OB and T2D. We focus on DNA methylation, the mechanism often associated with transgenerational persistence and for this reason frequently used in clinic, ecology and evolution studies [11].

## 2. T2D and OB: Risks, Prevalence, and Genetics

T2D and OB are complex diseases associated with numerous risk factors (cancer, retinopathy, nephropathy, myocardial infarction) [22,23,24,25]. The recent COVID-19 pandemic displayed one more risk factor: actually, patients with T2D or OB are at high risk of death in case of COVID-19 infection [19]. The prevalence of T2D and OB has rapidly increased during the last three decades, due to consumption of high-calorie foods, increased number of people aged 80 years or more and sedentary lifestyle. The rapid and global spread of these diseases indicates that environmental factors markedly contribute to these diseases. In addition, the majority of obese people are also diabetic, while minority groups are obese but not diabetic or with T2D and lean. Thus, T2D and OB patients are phenotypically heterogeneous [26]; in addition, the diagnosis of OB and T2D rely upon a single marker: BMI for OB and hyperglycemia for T2D. Thus, patients with the same disease may be largely different.

Genome-wide association studies (GWASs) have identified about 700 genes associated with OB [27] and 400 with T2D [28]. A large part of the genes associated with OB is regulated by the central nervous system [29], while those associated with T2D predominantly perturb the β cell function [30]. These results have encouraged the belief that a genetic test for the early diagnosis of patients at risk of developing OB or T2D would soon be available. However, given the complexity of these diseases, and the numerous associated risk factors, it is difficult that genes alone will ever be able to predict these diseases [26] (Figure 2).

Instead, preliminary results suggest that genes can contribute to better patient stratification. So far there have been five subgroups of T2D patients described. Two subgroups both display β-cell dysfunction, but one subgroup exhibits high and the other low levels of proinsulin (the prohormone precursor to insulin). Of the remaining three subgroups, one is characterized by obesity caused by insulin resistance, another by irregular body fat distribution, and the remaining one by an altered metabolism of liver fat [26]. Genes have also identified a highly interesting subgroup of patients predisposed to obesity but resistant to cardiometabolic disease (known as metabolically health obese phenotype) [31].

## 3. Epigenetics and OB

Global methylation indicates the level of methylcytosine expressed as a percent of total cytosine. The majority of studies—including a very large one [32]—did not find an association between OB and global methylation. Instead, two more studies reported an association between OB and global methylation [32,33,34]. Since many factors may influence global methylation (age, gender, alcohol, diet and many more), this approach was replaced with more specific methods.

Genome-wide studies investigate DNA methylation across a large number of genes. One of these studies reported that the DNA methylation level of peripheral blood leucocytes was higher in obese cases than in lean controls [35]. A later study detected five hyper-methylated sites in peripheral blood; three of these sites were located in the *HIF3A* gene that regulates the response to low oxygen. Interestingly, the same sites were highly methylated also in the adipose tissue [36]. A large study that included data from 10,000 blood samples detected 187 methylated sites associated with BMI. Using Mendelian randomization, the authors demonstrated that altered methylation—in the blood and the adipocytes—in the majority of cases results from obesity [37]. The DNA profile of isolated fat cells from women 2 years after a gastric bypass was compared with the profile of fat cells isolated from weight-matched but not obese women. Of the 8504 differently methylated CpG sites, 27% were associated with adipogenesis. This result explains the very high number of fat cells detected in obese and post-obese individuals [38].

Studies on methylated candidate genes focus on genes already known to be associated with OB. These studies have identified reduced methylation of tumor necrosis factor-alpha (*TNFα*) in peripheral blood [39], *PPARg* coactivator 1 alpha (*PGC1α*) in muscle [40], pyruvate dehydrogenase (*PDK4*) in muscle [40], and leptin in whole blood [41]. The more frequently confirmed association was that between *IGF2*/*H19* and adiposity [42].

## 4. Epigenetics of T2D

The first study [43] showed that the DNA methylation profiles of T2D patients and non-diabetic controls were different, suggesting that epigenetics might have a role in T2D. Later studies analyzed the methylation level of genes known to be associated with insulin resistance: *INS*, *PDX1* and *PPARGC1A*. Pancreatic islets from T2D patients displayed increased levels of DNA methylation and decreased expression of the above genes [43,44]. Dayed et al. [45] analyzed the methylation and transcription levels of genes in the pancreatic islets from T2D patients and non-diabetic controls. The authors detected 1649 CpG sites attributable to 853 genes. The 102 differently methylated genes—which included *CDKN1A*, *PDE7B*, *SEPT9*, and *EXOC3L2*—were differently methylated also in the islets of T2D patients. Functional analysis demonstrated that the above candidate genes affect insulin secretion, exocytosis in pancreatic β-cells, and glucagon secretion in α-cells (Figure 3).

In T2D, hepatic insulin resistance is associated with non-alcoholic fatty liver disease (NAFLD). To disclose the mechanism associating hepatic insulin resistance and NAFLD, the authors of this study [46] compared the methylome and transcriptome of livers from patients with T2D with the methylome and transcriptome of livers from individuals with normal plasma glucose levels. The livers from obese individuals displayed hypomethylation at a CpG site of the gene, encoding the platelet-derived growth factor α (PDGFA). PDGFA dimerizes forming the PDGF-AA homodimer. The livers of obese individuals—in addition to hypomethylation at the CpG site in the gene *PDGFA*—displayed overexpression of the same gene and increased synthesis of the PDGF-AA protein that contributes to insulin resistance through decreased expression of the insulin receptor. Interestingly, neutralization of the PDGF-AA excess with anti-PDGF-AA antibodies re-established hepatic insulin sensitivity and the associated NAFLD. The study underlines the therapeutic potential of epigenetic studies.

The case study that we now outline shows how epigenetics sometimes discerns what genetics misses. Transgenic mice overexpressing the *ped*/*pea*-*15* gene in the pancreatic β cells modulate insulin secretion [47]. Thus, this gene can actually be assumed associated with T2D. However, several independent case-control studies failed to detect this association. Instead, the same authors [48,49] demonstrated that diets causing obesity in mice and in humans alter the acetylation level of the histone H3 in the promoter of *ped*/*pea*-*15* and enhance the transcription of this gene. These studies explain the negative effect of diets on glucose tolerance and show that epigenetics is crucial to understand phenotypic differences characterizing single patients.

## 5. Interaction between Genetics and Epigenetics in T2D and OB

The identification of the single nucleotide polymorphism (SNP) that alters the melatonin receptor (*TCF7L2*)—a gene associated with T2D [50]—was made possible by combining epigenome annotation and genetic mapping. Pancreatic cells, enteroendocrine cells, and adipocytes all express this gene. The use of a humanoid mouse model overexpressing *TCF7L2* showed that the extra copy added to *TCF7L2* induces insulin resistance. Then, exploiting gene editing, the same authors could establish that one single nucleotide change in the risk allele rs7,903,146 of *TCF7L2* is sufficient to repress adipogenesis and produce hypertrophic cells. These results clearly show that only the comprehensive knowledge of the underlying genetic and epigenetic mechanisms of genes can lead to the full understanding of a disease.

The prevalent human gut microbiota phyla are *Bacteroidetes*, *Firmicutes*, and *Proteobacteria*. Microbiota and its metabolites influence genomic reprogramming [51]. *Firmicutes* produce butyrate, which modifies gene expression by histone modification [51]. Lipopolysaccharide (LPS)—also a microbial factor—induces inflammation, a risk factor for cardiovascular disease (CVD) [52]. The purpose of the study we are describing is to know if the host-microbiota interaction during the sensitive period of pregnancy poses to the host the risk of developing OB or other diseases later in life. Sequencing of DNA methylation from eight pregnant women (four with a microbiota enriched in *Bacteroidetes* and *Proteobacteria*; four with a microbiota enriched in *Firmicutes*) displayed differential methylation patterns: 245 genes displayed a lower level of methylation in the mothers with a higher level of *Firmicutes* than in those with higher levels of *Bacteroidetes* and *Proteobacteria* [47]. The epigenetically regulated genes included *USF1* (a regulator of fatty acid synthesis and lipogenesis), and *LMNA* (associated with CVD) [53]. These results agree with independent studies reporting that high levels of *Firmicutes* are associated with OB [54], T2D [55] and CVD [56]. The study clearly shows how microbiota, epigenetics, and disease risks are interdependent and identifies microbiota as a new target to prevent CVD.

Necrotizing enterocolitis (NEC) is an inflammatory bowel disease affecting premature infants. The disease—often lethal—develops in the absence of microbiota in the intestine of preterm infants, a condition that favors the infection by pathogens. However, a causal pathogen has not yet been isolated. Since a decrease in *Firmicutes* precedes the development of NEC [57], the authors of the case study that we are describing [58] speculated that the causal factor of the disease may be the absence of intestinal microbiota, rather than a pathogenic infection. This hypothesis finds support in the notion that epigenetic changes occurring during fetal life influence the composition of the intestinal microbiota [58]. Further—aware that dexamethasone often given to women in preterm labor to favor the lung maturation of the preterm infant can also alter the DNA methylation—the authors used a mouse model of prenatal exposure to dexamethasone to demonstrate that antenatal treatment with glucocorticoids alters the epigenome of offspring. Five candidate genes associated with inflammation displayed DNA methylation changes. Antenatal exposure to dexamethasone reduced also the *Clostridia* number in the gut of offspring compared to control offspring. *Clostridia* are essential to maintain the gut immune homeostasis [59]. This study and the previous one demonstrate that epigenetic changes occurring during the intrauterine life may alter the microbiota and predispose infants to diseases later in life. We add that a variety of dietary components and metabolites synthesized de novo by the host influence the epigenome and cause diseases. The identification of these factors is a huge challenge, but also the key to assess risk and stratify heterogeneous disease, leading the way to precision medicine.

## 6. Role of Aging in T2D and OB

T2D and OB display higher prevalence in older populations [60] and involve both men and women. In old patients, T2D is associated with typically old age comorbidities, such as urinary incontinence, sarcopenia and cognitive impairment. OB increases the risk of the above conditions, insulin resistance, and altered β cell function, which in turn promote T2D [61]. Aging is also characterized by an altered metabolism and production of reactive oxygen species (ROS), which damage cellular and mitochondrial proteins, lipids and DNA [62].

## 7. Intermittent Fasting, Epigenetics and Aging

While biological aging causes age-related diseases, such as metabolic, cardiovascular, and neurodegenerative diseases [63], there is growing evidence that intermittent fasting (IF) has anti-aging effects, as confirmed by several studies carried out in animal models and in humans [64]. IF (based on a fasting period between two meals) was shown to shift metabolism from lipid synthesis and fat storage to fat mobilization through fatty acid oxidation [65]. In addition, it increases autophagy, reduces inflammation, and modulates gut microbiota [66].

Obese women—who changed their diet from multiple daily meals to alternate-day energy restriction—reached a significant reduction of TNF-α and Il-6 levels [67]. The results observed with IF and the related approach of caloric restriction rely on two important properties of epigenetic markers: one is of being reversible; the second of being modulated by environmental factors, including the diet [68].

## 8. Discussion

In this section, we revisit the multiple roles of epigenetics and try to account for the seemingly contradictory results associated with epigenetics.

Worldwide, the overweight or obese subjects are about 1.5 billion [37]. Approximately, this is also the number of people potentially exposed to the risk of developing T2D and other complications associated with T2D. A large epigenome-wide association study showed that BMI—the canonical biomarker of OB—is associated with changes in DNA methylation (*p* < 10^−7^; *n* = 10,261 samples) [37]. This result demonstrates that altered DNA methylation is the consequence of adiposity, rather than the cause. This conclusion is in line with the independent evidence that in obese subjects, loss of weight leads to a partial remission of obesity-associated methylation sites [40].

Further, the same study [31] shows that 62—out of the detected 187 methylated sites—are associated with T2D (*p* < 2.7 × 10^−4^) and predict the future development of T2D. The strongest association detected in the study is with *ABCG1*, the gene regulating insulin secretion and cell function of pancreatic β-cells [69].

This outstanding study shows that genetics and epigenetics are complementary, rather than antagonistic: used together they offer the possibility to distinguish between association and causality. Even more important is the possibility to predict the development of T2D. Early detection of T2D is “vital” [26] since can reduce long-term complications, such as retinopathy: there is no cure for this disease, but the sooner the treatment starts, the better are the results [70]. Further, the chance of achieving reversal of T2D decreases with the increasing duration of the disease [71]. Early diagnosis may also prove very useful for the stratification of patients and personalized medicine [26].

Generally, genetics and lifestyle are the two factors called into question to explain metabolic diseases, T2D and OB in particular. However, there is convincing evidence that unfavorable conditions during intrauterine life (deficient or excessive nutrient intake by the mother [19]; microbiota enriched in *Firmicutes* [53]; prenatal exposure to dexamethasone) [58] can induce metabolic adaptations. These adaptations (insulin resistance, low rate of anabolic hormones, and deviation of glucose from ATP production to aerobic glycolysis to increase the production of glucose for vital organs) permit the survival of the fetus in unfavorable prenatal conditions. However, these adaptations come at the cost of predisposing the newborn baby—once adult—to T2D under conditions of abundant food. One can pose the question, is this the best solution that natural selection can find? The short answer is yes. Natural selection does not select for health, but only for reproductive success. If a gene implements reproduction—by any means—it will increase its frequency in the population [72]. Furthermore, T2D usually is not harmful before age forty, therefore the fact that someone will develop T2D after age forty cannot influence how many children he planned to have.

Epigenetics controls differentiation [73] development [74], and gene regulation [75]. Furthermore, integration of epigenetics and genetic data show us how environmental factors contribute to T2D [76] and OB [77]; locus-specific epigenetic targeting is expected to lead to new therapies against T2D and OB [50]. At the same time, epigenetics caused cancer [78], T2D, OB, and CVD [13]. How can we rationalize the contrasting roles of epigenetics?

Organisms look optimally designed but often are sub-optimal. Furthermore, the optimally and sub-optimally designed traits come all at a cost (tradeoff), which is often expressed in terms of disease [72]. The human body—as any living organism—has compromises, each providing an advantage and its corresponding tradeoff. The randomness of the evolutionary process contributes to the organization of genes in clusters: one cluster common to several diseases [79] or to the single disease [80,81] or multiple polymorphisms of the same gene for a single disease [82].

This last paragraph outlines the role of epigenetics in the context of evolution, beyond metabolic diseases. The concept of molecular evolution rests on genes and their random mutations, with the latter creating the genetic variation on which natural selection works [83]. The limit of this theory resides on the too low frequency of advantageous genetic mutations [83] and the too high frequency of the phenotypic mutation [84]. A unified theory of molecular evolution that includes genetics and epigenetics explains the above discordance and several facts that otherwise remain without a convincing explanation. (1) For example, the case of identical twins that share the same genotype but display discordant diseases [85]. (2) Diseases—such as T2D and OB—that have increased their frequency very fast to be attributable to gene mutations. (3) The hundreds of environmental contaminants that do not alter DNA frequency, but later in life display epimutations that predispose to diseases [86]. (4) The rapid evolution of certain traits [87]. Epigenetics explains all the above findings in the context of Darwinian evolution; in addition, it describes how the environment contributes to evolution.

## 9. Conclusions

In conclusion, how can we promote a better understanding of T2D and OB? Progress will require studying these diseases at population and individual levels. Population studies will identify interactions between genetics and epigenetics and differences between genes at the expression level and those between single SNP frequency. The studies at the individual level will clarify why people with the same disease respond differently to the same therapy and permit us to stratify patients into more homogeneous subgroups; the first step toward precision medicine.

## Figures and Tables

**Figure 1 biomedicines-09-00977-f001:**
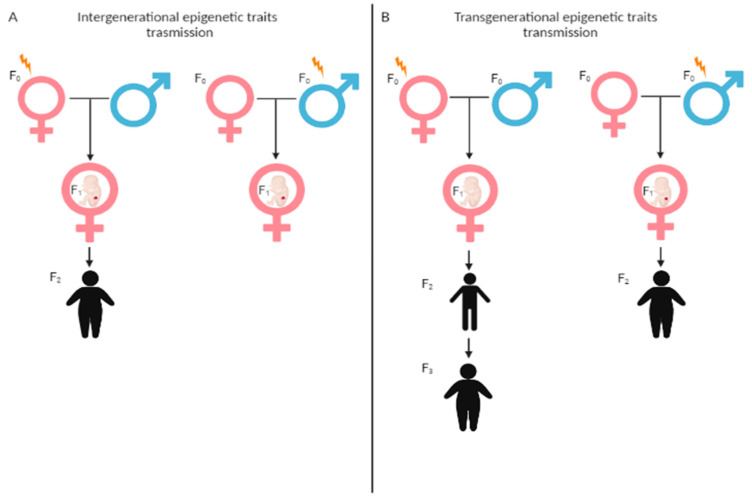
Intergenerational vs. Transgenerational inheritance. (**A**) Intergenerational inheritance occurs when exposure to the maternal womb influences the developing fetus and its germline, leading to an altered phenotype of the child. (**B**) Transgenerational inheritance occurs when exposure to the maternal womb influences the developing fetus and its germline and is transmitted to the F2 or F3 generation.

**Figure 2 biomedicines-09-00977-f002:**
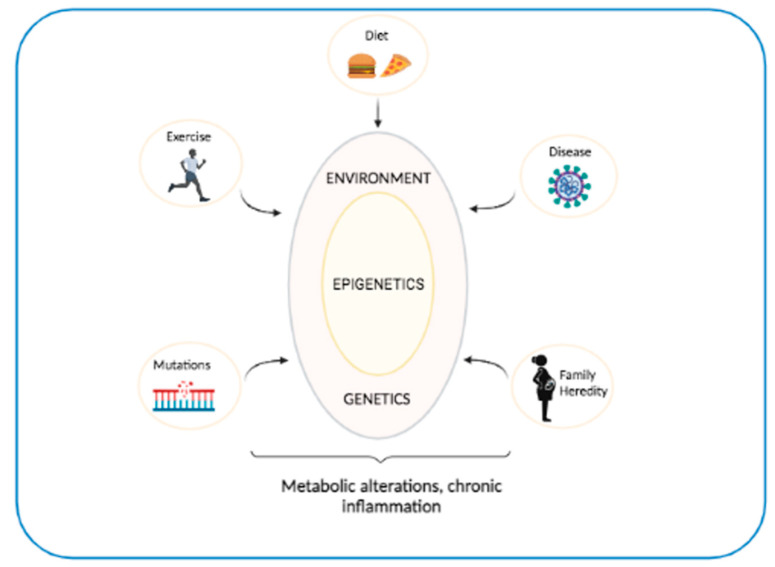
The role of genes and environment in T2D and OB. The interactions between environmental factors and genetic predisposition play a critical role in affecting metabolic processes and promoting chronic inflammation, which leads to severe diseases such as T2D and OB.

**Figure 3 biomedicines-09-00977-f003:**
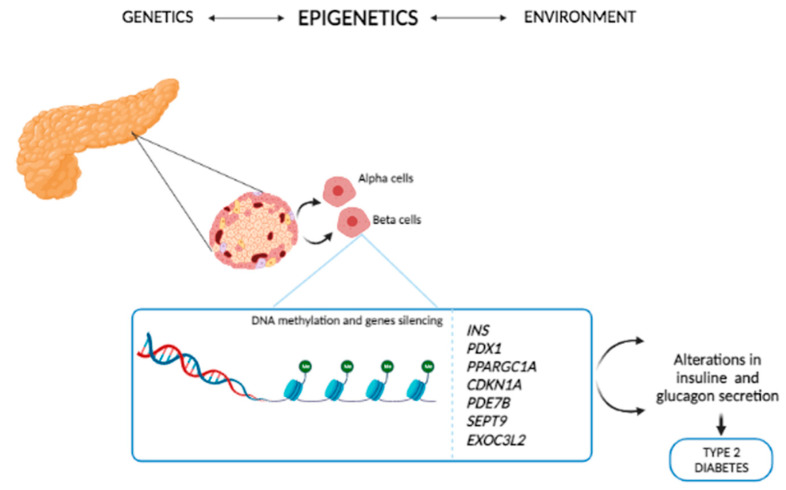
Effects of epigenetics in T2D gene regulation. The genes *CDKN1A*, *PDE7B*, *SEPT9*, *EXOC3L2*, *INS*, *PDX1* and *PPARGC1A* regulate insulin, exocytosis in pancreatic β cells and glucagon secretion in α cells in T2D patients.

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
