# Peer review of "Role of Epigenetics in Type 2 Diabetes and Obesity"

_biomedicines, 2021, doi:10.3390/biomedicines9080977_

Round 1

Reviewer 1 Report

Review for Biomedicines-1312789

It is very interesting to present a strategic implication for the treatment of Type 2 Diabetes and obesity through epigenetic level. In this paper, the authors suggested that epigenetic mapping is very important to better understand the phenotypic differences that plague a single patient. This present review was pretty well written and organized, but there are still some things that should to be corrected.

  1. In order to better understand the epigenetic changes that occur early in life due to insufficient intrauterine nutritional conditions, more references to relevant clinical papers should be added to Introduction (page 2, Line 8-14).
  2. The authors suggest that an excess or lack of nutrients during intrauterine lifespan may cause epigenetic changes increase OB, T2D in children adults. In this regard, have the authors considered a recent paper on the effects of "intermittent fasting" on obesity and diabetes? Intermittent fasting is an acquired factor that can control metabolic diseases, and research on it is receiving attention recently.
  3. On page 2 (line 31-34), obesity and type 2 diabetes are complex diseases associated with numerous risk factors, but the COVID19 pandemic does not make sense to incorporate them into epigenetic studies. Authors should provide additional references in this regard, or it is not recommended to address inlogical interpretations.
  4. In addition, a brief description of the epigenitic analysis method for the diagnosis of OB or T2D is required in the Introduction or discussion section.
  5. Lastly, it is interesting to note that the human gut microbiota and its metabolites influence genome reprogramming. According to numerous studies, it has been reported that the gut microbiome including Bacteroidetes, Firmicutes, and Proteobacteria, is out of balance with aging. Aging is also an unavoidable event, which increases the risk of type 2 diabetes or obesity. Since aging may be a good tool to explain its epigenetic role to understand T2D/OB, authors need to consider and discribe in this review. (page 5, Line 23~)

Author Response

It is very interesting to present a strategic implication for the treatment of Type 2 Diabetes and obesity through epigenetic level. In this paper, the authors suggested that epigenetic mapping is very important to better understand the phenotypic differences that plague a single patient. This present review was pretty well written and organized, but there are still some things that should to be corrected.

  1. In order to better understand the epigenetic changes that occur early in life due to insufficient intrauterine nutritional conditions, more references to relevant clinical papers should be added to Introduction (page 2, Line 8-14).

As kindly suggested, new references have been added: N E J Med 359;61 – Nature 430; 419 – Lancet 1; 1077 – BMJ 307; 1519. Please, see page 2, line 23.

2. The authors suggest that an excess or lack of nutrients during intrauterine lifespan may cause epigenetic changes increase OB, T2D in children adults. In this regard, have the authors considered a recent paper on the effects of "intermittent fasting" on obesity and diabetes? Intermittent fasting is an acquired factor that can control metabolic diseases, and research on it is receiving attention recently.

Please, see page 6, lines 25-39.

3. On page 2 (line 31-34), obesity and type 2 diabetes are complex diseases associated with numerous risk factors, but the COVID19 pandemic does not make sense to incorporate them into epigenetic studies. Authors should provide additional references in this regard, or it is not recommended to address inlogical interpretations.

As kindly suggested, the comment on COVID-19 has been omitted. Please, see page 2, lines 31-34.

4. In addition, a brief description of the epigenitic analysis method for the diagnosis of OB or T2D is required in the Introduction or discussion section.

A brief description of the epigenetics methods has been added in the Introduction section. Please, see page 1 (lines 36-44) and page 2 (lines 1-2). 

5. Lastly, it is interesting to note that the human gut microbiota and its metabolites influence genome reprogramming. According to numerous studies, it has been reported that the gut microbiome including Bacteroidetes, Firmicutes, and Proteobacteria, is out of balance with aging. Aging is also an unavoidable event, which increases the risk of type 2 diabetes or obesity. Since aging may be a good tool to explain its epigenetic role to understand T2D/OB, authors need to consider and discribe in this review. (page 5, Line 23~)

Thank you for your suggestion. The epigenetic role of aging T2D and OB is described apart. Please, see page 7, lines 1-9. 

Reviewer 2 Report

27th July, 2021

Review of the Manuscript ID  biomedicines-1312789, by R. Capparelli and D. Iannelli, entitled: “Role of Epigenetics in Type 2 Diabetes and Obesity” that is intended to be published as the Review in Biomedicines

(separate Microsoft Word file as Reviewer Attachment for Manuscript ID biomedicines-1312789 Biomedicines 27th July 2021 that includes Comments to the Authors is also uploaded)

Taking into consideration research highlight, contribution of the Authors to the progress in the research field, thorough manner of data presentation, perfectly writing in English, abundance of Figures (diligent graphic documentation), the quality of this paper deserves praise and merits my support. The Authors have received the high scores from me for the originality, importance of the work and the scientific value of their paper. In my opinion, the current paper provides insightful interpretation of topical trends in the elucidation of epigenetic and molecular mechanisms underlying etiology and etiopathogenesis of the most common lifestyle/civilization diseases such as: type 2 diabetes and obesity. For all these reasons, I strongly recommend the Editorial Board to allow for publication of this very interesting paper in Biomedicines, after the minor revision of the manuscript will have been completed by the Authors and provided that the Authors are ready to consider all the Reviewer comments indicated below:

1) There is a lack of the separate Conclusions and Abbreviations sections in the paper. Therefore, these sections should have been added by the Authors to the manuscript.

2) The References section has to be prepared in the format compatible with the requirements of Biomedicines.

General Comment of the Reviewer:

Before the manuscript will have been accepted for publication in Biomedicines, it requires the minor revision (according to all the remarks and suggestions of the Reviewer).

Author Response

Taking into consideration research highlight, contribution of the Authors to the progress in the research field, thorough manner of data presentation, perfectly writing in English, abundance of Figures (diligent graphic documentation), the quality of this paper deserves praise and merits my support. The Authors have received the high scores from me for the originality, importance of the work and the scientific value of their paper. In my opinion, the current paper provides insightful interpretation of topical trends in the elucidation of epigenetic and molecular mechanisms underlying etiology and etiopathogenesis of the most common lifestyle/civilization diseases such as: type 2 diabetes and obesity. For all these reasons, I strongly recommend the Editorial Board to allow for publication of this very interesting paper in Biomedicines, after the minor revision of the manuscript will have been completed by the Authors and provided that the Authors are ready to consider all the Reviewer comments indicated below:

  • There is a lack of the separate Conclusions and Abbreviations sections in the paper. Therefore, these sections should have been added by the Authors to the manuscript.

As kindly suggested, Conclusions and Abbreviation sections have been added to the manuscript.

  • The References section has to be prepared in the format compatible with the requirements of Biomedicines.

As kindly suggested, references have been prepared in the format compatible with the requirements of Biomedicines.  

Round 2

Reviewer 1 Report

The authors had corrected the draft and confirm the influence between T2D/OB epigenetics and intermittent fasting/aging. The writing was presented properly in a clear and concise form. The suggestions were considered by the authors and contributed further to scientific quality. Therefore, the manuscript was revised adequately and can be accepted for publication in Biomedicines.